# Genetic–Geographic–Chemical Framework of *Polyporus umbellatus* Reveals Lineage-Specific Chemotypes for Elite Medicinal Line Breeding

**DOI:** 10.3390/jof12010039

**Published:** 2026-01-03

**Authors:** Youyan Liu, Shoujian Li, Liu Liu, Bing Li, Shunxing Guo

**Affiliations:** 1The Institute of Medicinal Plant Development, Chinese Academy of Medical Sciences & Peking Union Medical College, Beijing 100193, China13756298450@163.com (S.L.);; 2State Key Laboratory for Quality Ensurance and Sustainable Use of Dao-di Herbs, Beijing 100193, China

**Keywords:** *Polyporus umbellatus*, genetic diversity, metabolite profiling, whole-genome sequencing, phylogenetics analysis

## Abstract

*Polyporus umbellatus* is a valuable fungus with both dietary and medicinal applications. However, heterogeneous germplasm and chemical variability constrain its sustainable use. To elucidate the drivers of this variation, whole-genome resequencing and metabolic profiling were integrated. Genome-wide analysis of representative accessions revealed six distinct genetic clusters across China, identifying the Qinling–Daba Mountains as a putative center of diversity. Population analysis indicated severe genetic erosion with significant heterozygote deficits, likely driven by inbreeding and long-term clonal propagation. Multivariate analysis demonstrated that genetic lineage, rather than traditional commercial morphotypes (Zhushiling and Jishiling), is the primary determinant of metabolite accumulation. Specific lineages were identified as superior germplasm candidates: Group 2 consistently exhibited the highest genetic potential for accumulating steroids, whereas Group 4 attained the highest polysaccharide yield. Although the global genetic–chemical correlation was weak, implying environmental plasticity, the distinct clustering of superior lineages confirms that core accumulation patterns are genetically canalized. These findings advocate for shifting quality control from morphological grading to molecular-assisted selection. Ultimately, this framework provides an evidence-based foundation for urgent in situ conservation to restore genetic diversity and facilitates precision breeding of high-efficacy cultivars.

## 1. Introduction

*Polyporus umbellatus* (Pers.) Fries, a medicinal basidiomycete within the family Polyporaceae, is widely distributed in temperate oak forests [1]. Its sexual life cycle comprises four distinct stages: basidiospores, mycelia, sclerotia, and fruiting bodies [2]. The epigeous fruiting bodies are consumed as food [3], while subterranean sclerotia have been utilized as traditional medicine for over 2500 years [2] and remain officially listed in the Chinese Pharmacopoeia [4]. Beyond widespread prescriptions of *P. umbellatus* preparations for nephropathies in Asia, the fungus is increasingly incorporated into nutraceuticals and cosmeceuticals in Europe and North America [5,6]. Polysaccharides from *P. umbellatus* (PUPs) exhibit antineoplastic, hepatoprotective, and immunomodulatory activities with negligible toxicity [1,7,8,9,10]. Consequently, PUPs are approved in China for the treatment of chronic hepatitis B and as oncological adjuvant therapy [1]. Furthermore, sterols derived from the fungus display diverse therapeutic effects, including diuretic, nephroprotective, and anti-inflammatory properties [11,12,13,14,15,16,17,18]. These multifaceted bioactivities underscore the significant medicinal and commercial value of *P. umbellatus* [16,19]. However, escalating demand for medicinal raw materials has resulted in the overexploitation of wild resources, severely threatening wild populations. Consequently, the species is now legally protected in China and classified as endangered or vulnerable in various European regions [20,21,22,23]. Although artificial cultivation was introduced in China over 45 years ago to mitigate resource depletion, long-term farming has caused genetic admixture and deterioration of quality traits, hindering the sustainable development of the industry [2,24].

Despite the economic and medicinal importance of *P. umbellatus*, its genetic background remains poorly characterized. Previous studies utilizing markers such as SRAP, ISSR, and EST-SSR [25,26,27,28] were constrained by limited sample sizes and low resolution [25,29]. Moreover, a validated protocol for germplasm identification does not currently exist. Commercially, sclerotia are morphologically categorized as either Zhushiling (large, sparsely branched, and smooth) or Jishiling (small, densely branched, and wrinkled) [30]. Although Zhushiling is widely regarded as superior [31], systematic genetic and chemical evidence supporting this quality distinction is currently lacking [32,33,34]. Quantifying intraspecific genetic variation is therefore imperative to ensure a high-quality supply of medicinal raw materials. This variation influences both species viability [35] and batch-to-batch consistency in traditional Chinese medicines [36]. Furthermore, genomic characterization will underpin the development of elite cultivars and determine which genetic lineages yield sclerotia suitable for pharmaceutical applications [30].

Recent advances in molecular systematics provide robust tools to effectively address these knowledge gaps. Specifically, SSR markers, valued for their high polymorphism and technical maturity, represent a standard for mesoscale population-genetic studies [37]. Additionally, high-throughput SNP profiling via next-generation sequencing enables simultaneous sampling of thousands to millions of genomic loci, providing necessary resolution for fine-scale inference of population structure and demographic history [38]. Thus, this study integrates genome-wide SSRs, SNPs, and ITS/LSU barcodes with chemical phenotyping to dissect genetic diversity and population structure in *P. umbellatus*. Specifically, this study aims to: (1) explicitly test genetic differentiation between Zhushiling and Jishiling morphotypes; (2) elucidate how geographic variation shapes phenotypic and chemical divergence; and (3) construct a genetic-geographic-chemical evaluation framework. These findings will establish a foundation for conservation strategies, elite-strain selection, and sustainable utilization of this medicinally important fungus.

## 2. Materials and Methods

### 2.1. Sampling and Genomic DNA Extraction

A total of 53 *Polyporus umbellatus* samples were collected from 9 provinces across China. Based on sclerotial morphology, samples were classified into 2 morphotypes: Jishiling (*n* = 23) and Zhushiling (*n* = 27). Mycological images of collected samples are presented in Appendix A. The remaining three samples consisted of *P. umbellatus* strains and fruiting bodies, whose sclerotial types could not be determined. Additionally, 42 ITS and LSU sequences from 12 Chinese provinces [30] were integrated for phylogenetic analysis. Consequently, the dataset comprised 95 samples, representing seven major mountain populations across China [39] (Appendix A). Genomic DNA was extracted from fresh sclerotia using a modified CTAB method [40]. DNA quality and concentration were verified using a NanoDrop One spectrophotometer (NanoDrop Technology, Wilmington, DE, USA). Voucher specimens are deposited at the Institute of Medicinal Plant Development (IMPLAD), Beijing, China.

### 2.2. Genomic SSR Mining and Marker Development

Microsatellite loci (SSRs) were identified within the *P. umbellatus* genome [2] using MicroSAtellite identification tool (MISA, v1.0) [41]. SSRs were filtered according to criteria outlined in Appendix A. To maximize polymorphic potential, a targeted strategy prioritizing compound SSRs and loci characterized by high repeat numbers was employed [42,43]. Primers were designed using Primer3 v2.3.7 [44], adhering to stringent parameters regarding melting temperature (Tm), GC content, and secondary structure (detailed parameters provided in Appendix A). Initial validation of marker polymorphism was performed by amplifying genomic DNA from six representative germplasm accessions and visualizing the PCR products via 8% native polyacrylamide gel electrophoresis (PAGE). Ultimately, twenty primer pairs exhibiting distinct amplification bands and significant polymorphism were selected for subsequent population genetic analysis.

### 2.3. SSR Genotyping and Diversity Analysis

The 53 newly collected accessions were genotyped using the 20 selected polymorphic SSR markers. Forward primers were fluorescently labelled (6-FAM, HEX, ROX, or TAMRA) at the 5′ end. PCR amplifications were conducted according to established protocols [28]. with specific reaction mixtures and thermal cycling conditions detailed in Appendix A. PCR products were analyzed using an ABI 3730XL Genetic Analyzer (Applied Biosystems, Foster City, CA, USA) and scored using GeneMarker v2.20. Gel images were examined to determine PCR band presence (1) or absence (0). Genetic diversity parameters and Analysis of Molecular Variance (AMOVA) were calculated using GenALEx 6.51 [45] and PowerMarker 3.25 [46]. Phylogenetic relationships were inferred using the UPGMA method based on genetic similarity coefficients via NTSYSpc v2.10e. Population structure was analyzed with STRUCTURE v2.3.4 [47] performing 20 independent runs for *K* = 1–10 (burn-in: 10,000; MCMC: 500,000 iterations). The optimal number of clusters was determined using the ΔK method on the Structure Selector platform.

### 2.4. SNP Identification and Population Genetic Structure Analysis

High-throughput genome resequencing was performed on the DNBSEQ-T7 platform (BGI, Shenzhen, China). Raw reads were filtered to remove adapters and low-quality bases using standard bioinformatic pipelines (detailed criteria in Appendix A). Clean reads were mapped to the *P. umbellatus* reference genome [2] using bwa-mem2 mem-t4-M v2.2 [48]. Resulting SAM files were converted to BAM format, sorted, and indexed using samtools 1.9 [49]. Variant calling was conducted with GATK HaplotypeCaller [50], and variants were annotated with SnpEff [51]. Population structure based on genome-wide SNPs was analyzed using three complementary approaches: ADMIXTURE v1.22 [52] (with cross-validation to determine optimal K), Principal Component Analysis (PCA), and Maximum Likelihood (ML) phylogenetic reconstruction.

### 2.5. Phylogenetic Study of Chinese P. umbellatus Populations

To place populations in a broader phylogeographic context, the ITS and 28S rRNA (LSU) regions were amplified using universal primer pairs (ITS1/ITS4 and LROR/LR7). PCR amplification followed standard cycling conditions (Appendix A). A combined dataset of 95 concatenated ITS-LSU sequences (53 newly obtained sequences and 42 references) was used to construct a Maximum Likelihood tree in MEGA7 [53] employing the Kimura 2-parameter model with 1000 bootstrap replicates. Correlations between genetic and geographical distances were assessed via Mantel tests (999 permutations) using the vegan package in R v4.2.1 [36].

### 2.6. Determination of P. umbellatus polysaccharide and Three Steroids

Quantitative analysis of four key medicinal components—total polysaccharide, ergosterol, polyporusterone A, and polyporusterone B—was conducted on sclerotia from 47 *P. umbellatus* samples collected across 10 provinces in China. Six laboratory-preserved strains were excluded from this analysis due to the absence of sclerotia; detailed information regarding these strains is provided in Appendix A. Total polysaccharide content was quantified using the phenol-sulfuric acid method. Briefly, each pulverized sample (2.0 g) underwent ultrasonic extraction for 1 h at room temperature in 40 mL ultrapure water. After extraction, the resulting supernatant was collected, and its absorbance at 490 nm was recorded against a standard calibration curve established with D-glucose. Ergosterol, polyporusterone A, and polyporusterone B were quantified by employing High-Performance Liquid Chromatography (HPLC). For sterol extraction, pulverized sclerotia (1.0 g) were sonicated in methanol (10 mL) for 1 h. The final extracts were adjusted for any weight reduction by supplementing methanol, then filtered using a 0.45 µm filtration membrane prior to analysis. An XSelect CSH C18 chromatographic column (250 mm × 4.6 mm, 5 µm) connected to a Waters HPLC system facilitated sterol separation. The chromatographic conditions involved gradient elution using water, acetonitrile, and methanol, with specific gradient details listed in Appendix A. Commercially sourced standard compounds of high purity (>98%) were utilized for comparison, with additional supplier information provided in Appendix A.

Statistical analyses were conducted using SPSS version 27.0. Normality was evaluated by the Shapiro–Wilk test. Parametric data were analysed by one-way ANOVA followed by Tukey’s HSD post hoc test (η^2^) or independent-sample *t*-test. Non-parametric data were assessed using the Kruskal–Wallis H test with Bonferroni-corrected Dunn’s post hoc test (ϵ^2^) or the Mann–Whitney U test (*r*). Principal Component Analysis (PCA) was conducted in OriginPro 2022 based on the correlation matrix. Statistical significance was defined as a *p*-value < 0.05. For subsequent analyses, chemical data were standardized through Z-score normalization [54]. Hierarchical clustering analysis (Ward’s D2 method, Euclidean distance) was executed using OmicStudio(LC-Bio Technologies, Hangzhou, China).

## 3. Results

### 3.1. Genomic SSR Characterization and Marker Screening

In total, 2407 genomic SSRs (gSSRs) were identified in the *P. umbellatus* genome. The most common repeat motifs were dinucleotide (825; 34.28%) and trinucleotide (947; 39.34%) repeats. Among dinucleotide motifs, AG/CT was the most frequent (15.54%), while ACC/GGT was predominant (7.35%) among trinucleotide motifs. Appendix A illustrates the frequency and distribution of SSR types identified in the genome.

Based on the flanking sequences of the 2407 gSSR loci, 11,035 primer pairs were initially designed (Appendix A). Using a targeted approach focused on highly polymorphic regions, fifty gSSR primer pairs were selected for screening by 8% native PAGE, together with eight EST-SSR primers reported previously [28]. Ultimately, 20 primer pairs (6 EST-SSR and 14 gSSR) were selected based on their ability to consistently produce clear, prominent bands and high polymorphism. Detailed information and amplification patterns of these selected primers are shown in Appendix A and Table 1. Genotyping analysis performed on 53 *P. umbellatus* germplasms by capillary electrophoresis further confirmed their high discriminatory capacity (Figure 1).

Further genetic analysis demonstrated that these markers were suitable for assessing the genetic diversity and population structure of *P. umbellatus* (Table 2). The Polymorphism Information Content (PIC), an essential measure reflecting a marker’s ability to detect polymorphism, ranged from 0.19 (F21) to 0.78 (F46). Apart from primer F21, the other 19 primer pairs were classified as medium- to high-polymorphic markers (PIC > 0.5) and designated as core primers [55,56]. Notably, primers F46, F16, and PUF35 showed exceptionally high PIC values (0.78, 0.77, and 0.77, respectively). Observed heterozygosity (*H*_o_) ranged from 0 to 0.62, whereas expected heterozygosity (*H*_e_) ranged from 0.21 to 0.81. The consistently higher *H*_e_ compared to *H*_o_ indicated a deficit of heterozygotes within the sampled populations, suggesting the necessity for further exploration of the population’s genetic structure.

### 3.2. Genomic SNP Sequencing and Variant Identification

After stringent quality control and filtering, a total of 47 genomic DNA libraries from *P. umbellatus* were constructed, generating 163.50 Gbp of clean data, with an average Q30 value of 94.44%. Alignment to the reference genome yielded an average alignment rate of 89.28%, a mean sequencing depth of 38X, and genome coverage of 74.96%. Variant calling identified 1,026,121 high-quality variant sites, including 784,644 SNPs and 241,477 indels (Appendix A). SNPs (Ts/Tv = 3.35) predominantly occurred in CDS regions (31.49%) and intergenic regions (21.08%) (Appendix A). Within coding sequences, non-synonymous and synonymous SNPs accounted for 44.98% and 54.38%, respectively. Indels (96,591 insertions; 144,886 deletions) primarily appeared in 5 kb downstream (24.98%) and upstream (24.14%) regions (Appendix A). Within coding regions, indels mainly induced frameshift mutations (43.43%), followed by codon deletions (20.12%). A Circos plot (Appendix A) illustrated a positive correlation between SNP and indel densities, suggesting common genomic regions prone to mutations.

### 3.3. Population Clustering and Structural Congruence

Principal Coordinate Analysis (PCoA) and Principal Component Analysis (PCA) initially characterized population stratification. For SSR markers, the first two principal coordinates explained 49.84% of the cumulative variance (Figure 2A). SSR-based PCoA revealed a complex genetic structure; Group 1 formed a clearly defined cluster, whereas groups 2–6 substantially overlapped in the upper-right quadrant, suggesting possible gene flow or conserved genetic backgrounds among these lineages. Conversely, genome-wide SNP PCA (Figure 2B) provided clearer differentiation. The first two principal components, explaining 20.17% variance, distinctly separated the germplasms. Groups 1, 3, 4, 5, and 6 formed tight clusters, reflecting genetic uniformity. However, Group 2 exhibited considerable scatter, indicating higher internal genetic divergence. To explore this variability, a separate 3D PCA focused on Group 2 was performed (Figure 2C). This additional analysis confirmed significant internal variation among Group 2 (e.g., samples A11, A31), highlighting its status as the most genetically diverse subgroup in this collection.

To clarify population structure, model-based clustering was conducted using marker-specific algorithms. Both the Δ*K* method (for SSRs) and cross-validation error minimization (for SNPs) consistently determined *K* = 6 as the optimal number of genetic subgroups (Appendix A). Population structure plots (Figure 3) indicated that most germplasms exhibited high membership coefficients (*Q* > 0.8) within distinct clusters, reflecting defined genetic lineages with limited historical admixture. The SSR-based analysis partitioned the 53 germplasms into six discrete subgroups (Figure 3A), fully congruent with the SNP-based analysis results from the 47 sequenced accessions (Figure 3B). Nonetheless, the SNP-based analysis provided enhanced resolution regarding gene flow and introgression events. While most groups showed genetic homogeneity, specific admixture events were noted: Group 1 primarily comprised genotype GB1, but accessions like A13 and A14 (from Lushi, Henan) contained genotype GB3. Their exclusive assignment to GB3 in ADMIXTURE suggested reciprocal gene flow within this cluster. Accessions A26 and A27 (from Nujiang, Yunnan) formed a distinct subgroup (Group 3) but shared GB4 and GB6 genetic components with Groups 4 and 6, respectively. This genetic exchange explains the close phylogenetic proximity of Groups 3 and 4. Similarly, the presence of GB1 in A11 (Liuba, Shaanxi) and GB5 in A31 (Wangcang, Sichuan) within Group 2 accounts for their genetic isolation from the other three samples within the same group, as depicted by sub-PCA (Figure 2C).

### 3.4. Characterization of Six Genetic Lineages

Clustering analysis of the SSR data, using a genetic similarity threshold of 0.30, divided the 53 germplasms into six distinct clusters (Figure 4). This classification agreed with the topology of the SNP-based maximum likelihood (ML) phylogenetic tree (Figure 5). A significant correlation emerged between clustering outcomes and the geographic origins of germplasms, as individuals within the same cluster usually shared similar geographic distributions. Group 1 (*n* = 19, green) predominantly included germplasms from the Qinba Mountain population. Group 2 (*n* = 6, blue) consisted of germplasms from both Qinba Mountain and Changbai Mountain populations. Groups 3 (gray) and 4 (pink) exclusively contained germplasms from the Hengduan Mountain population. Group 5 (*n* = 11, yellow) mainly represented the Wumeng Mountain population, while Group 6 (*n* = 9, red) primarily originated from Changbai Mountain. Furthermore, germplasms grouped within the same cluster generally exhibited consistent morphological characteristics. For example, Groups 1, 3, and 4 primarily included the Zhushiling morphotype, whereas Groups 5 and 6 predominantly comprised the Jishiling morphotype.

The Analysis of Molecular Variance (AMOVA), employing the six genetic clusters (*K* = 6) identified by STRUCTURE as grouping variables, demonstrated highly significant genetic variation (*p* < 0.001) both among and within lineages. Most genetic variation (57%) occurred among individuals within populations, with a smaller portion (28%) within individuals, and the remaining 15% attributed to inter-population differences (Table 3).

### 3.5. Genetic Diversity Analysis

Genetic diversity analysis based on SSR markers revealed clear differences among the clusters (Table 4). Group 1 exhibited the highest genetic diversity, showing superior values across nearly all key metrics. This finding suggests that Group 1 should receive priority in conservation strategies. In contrast, Group 6 exhibited the lowest diversity, with the smallest observed heterozygosity (*H*_o_ = 0.17) and PIC (0.10). Group 3 was excluded due to insufficient sample size (*n* = 2). Except for Group 1, which showed an inbreeding coefficient (*F*_IS_) below 0.5, all other groups and both morphotypes of *P. umbellatus* exhibited *F*_IS_ values exceeding 0.5, indicating significant inbreeding within these populations [57].

A comparison of genetic diversity parameters among four mountain populations revealed distinctive patterns (Table 5). The Qinba Mountain population displayed the highest genetic diversity, leading in several metrics, including *N*_a_ (4), *I* (0.98), *H*_o_ (0.29), and PIC (0.41). These results imply the presence of a large and diverse allelic resource pool. Conversely, the Wumeng Mountain population showed the lowest genetic diversity in nearly all parameters, having the lowest *N*_e_ (2.08), *I* (0.80), and PIC (0.17). Its *N*_a_ (3) and *H*_o_ (0.14) were among the lowest, suggesting the presence of a highly restricted genetic resource. Moreover, regarding the two commercially recognized morphotypes (Table 6), the Zhushiling type showed higher genetic diversity than the Jishiling type. This conclusion is supported by elevated values across all six diversity parameters, indicating a richer and more diverse gene pool within the Zhushiling population.

### 3.6. Phylogenetic Study of Chinese P. umbellatus Populations

A phylogenetic tree was generated using the maximum likelihood (ML) method based on concatenated ITS and LSU sequences from 95 *P. umbellatus* germplasms collected across 14 provinces (Figure 6 and Figure 7). The resulting topology was highly consistent with patterns inferred from SSR and SNP markers, confirming the stability of the identified population structure. The absence of new major clades, even with expanded sampling, further validated the completeness of the current germplasm dataset. The ML tree resolved six well-supported clades, each exhibiting strong geographic associations. Group 1 (*n* = 46, green) consisted mainly of accessions from the Qinba Mountain and Taihang Mountain populations. Group 2 (*n* = 6, blue) included germplasms from both Qinba Mountain and Changbai Mountain regions. Group 3 (*n* = 2, gray) comprised two distinct Zhushiling accessions from Nujiang, Yunnan, belonging to the Hengduan Mountain population. Group 4 (*n* = 8, pink) contained three samples from Qinghai (Qilian Mountain population) together with Zhushiling individuals from the Hengduan Mountain region. Group 5 (*n* = 14, yellow) included three Jishiling accessions from Hanyin, Shaanxi, clustering with Jishiling individuals from the Wumeng Mountain population. Group 6 (*n* = 19, red) was composed almost entirely of accessions from the Changbai Mountain population.

To assess the influence of geographic distance on genetic differentiation, a Mantel test with 9999 permutations was performed. A significant positive correlation was detected between geographic and genetic distances (*r* = 0.3761, *p* = 0.0001), supporting an isolation-by-distance (IBD) pattern [58]. Genetic distances were calculated using the Kimura 2-Parameter model (Appendix A). This result was further corroborated by the strong concordance between genetic groupings and the geographic distribution of samples (Figure 7). For example, germplasms from the Qinba Mountain and Taihang Mountain regions formed a coherent genetic cluster, reflecting their spatial proximity. Likewise, accessions from the Changbai Mountain and Lesser Khingan Mountain regions clustered together, suggesting substantial genetic relatedness within this northeastern lineage.

### 3.7. Quantitative Analysis and Variation in Target Components

Quantitative analysis of four medicinal components (polysaccharide, ergosterol, polyporusterone A, and polyporusterone B) was conducted in 47 samples, revealing substantial variation in their accumulation (Appendix A). Hierarchical clustering analysis (HCA) categorized the samples into five clusters based on these analytes (Appendix A). Cluster I featured high concentrations of polyporusterones A and B. Specifically, the highest concentration of polyporusterone A (106.14 μg/g) appeared in the Liubao, Shaanxi sample, whereas polyporusterone B reached a maximum (172.58 μg/g) in the Wangcang, Sichuan sample. Notably, one sample from Wangqing, Jilin showed elevated levels of all four components: polyporusterone A (88.61 μg/g), polyporusterone B (144.49 μg/g), ergosterol (1933.31 μg/g), and polysaccharide (28.99 mg/g). Cluster II generally exhibited lower concentrations of all components. Cluster III displayed moderate polysaccharide levels but low ergosterol concentrations. Cluster IV was characterized by high levels of both ergosterol and polysaccharides, with a sample from Baishan, Jilin having the highest recorded values (2170.27 μg/g ergosterol; 33.79 mg/g polysaccharide). Cluster V samples exhibited moderate ergosterol but low polysaccharide concentrations.

### 3.8. Multivariate Analysis Identifies Genetic Lineage as the Main Factor Influencing Component Variation

To explore the drivers of chemical variability, Mantel tests were performed. The results indicated a weak yet significant positive correlation between genetic and chemical distances (*r* = 0.09, *p* = 0.03; Appendix A). However, no significant correlation emerged with geographical distance (*r* = 0.08, *p* = 0.20). PCA depicted the overall accumulation patterns, with the first two principal components explaining 76.7% of the variance, indicating strong dataset representation. The PCA biplot highlighted divergence among genetic lineages (Figure 8A). Polyporusterones A and B differentiated Group 2 (cyan), whereas polysaccharide and ergosterol defined Group 4 (magenta). Groups 1, 5, and 6 formed a central cluster, suggesting intermediate chemical profiles.

Multivariate patterns were confirmed through comparisons of the four components across five genetic clusters (Group 3, *n* = 2, excluded) (Appendix A). For polysaccharides (satisfying normality assumptions and homogeneity assumptions), one-way ANOVA (*F* (4, 39) = 2.74, *p =* 0.042, η^2^ = 0.22) revealed significant differences among groups. Tukey’s HSD post hoc tests specifically indicated that Group 4 had the highest concentration (24.23 ± 7.38 mg g^−1^), which was 2.0-fold higher than the levels observed in Group 6 (12.05 ± 4.99 mg g^−1^, *p* = 0.025) (Figure 9B). For the non-normally distributed Polyporusterone A and B (Figure 9C,D), the Kruskal–Wallis H test confirmed profound inter-cluster divergence consistent with the PCA observations. Polyporusterone A exhibited the most prominent differences (χ^2^ (4) = 27.38, *p* < 0.001, ϵ^2^ = 0.60), with Groups 1 and 2 significantly surpassing Groups 4 and 6 (Dunn’s post hoc tests, all *P*_adj_ < 0.01). Polyporusterone B also showed significant inter-group differences (χ^2^ (4) = 16.71, *p* = 0.002, ϵ^2^ = 0.33), with Group 2 significantly exceeding Groups 6 (*P*_adj_ = 0.004) and 4 (*P*_adj_ = 0.046). Ergosterol, despite influencing PCA separation of Group 4, did not differ significantly across groups (*F* (4, 39) = 0.41, *p* = 0.804).

In contrast, mapping morphotypes (Zhushiling vs. Jishiling) onto the PCA indicated considerable overlap (Figure 8B). Mann–Whitney U tests confirmed weak differentiation, detecting significant differences only for polyporusterone A (*U* = 178.50, *z* = −2.09, *p* = 0.037), Zhushiling (*n* = 22, median 27.24 μg g^−1^) exhibited concentrations 1.28-fold higher than Jishiling (*n* = 25, median 21.27 μg g^−1^), with a medium effect size (*r* = 0.30). Collectively, these multivariate and univariate analyses confirmed that genetic lineage predominantly drives chemical variation in *P. umbellatus*, while morphotype classification contributes minimally.

## 4. Discussion

To address the historical challenges of high germplasm heterogeneity and weak phenotype–genotype associations in *Polyporus umbellatus*, this study integrated whole-genome resequencing with quantitative analysis of medicinal metabolites. This multidimensional strategy establishes a high-resolution genetic framework for interpreting genetic, geographic, and chemical relationships, thereby providing a robust foundation for conservation and breeding efforts.

### 4.1. High-Resolution Genetic Structure and the Putative Diversity Center of P. umbellatus

This study presents the first comparative use of whole-genome-derived SSRs and resequencing-based SNPs in *P. umbellatus*. Although both marker types were informative, the dataset of 784,644 high-quality SNPs offered markedly higher resolution than the 19 core SSR markers and previously applied marker systems (e.g., SRAP, ISSR) [25,26,28]. The high-resolution SNP data strongly support the Qinling–Daba Mountains as a putative diversity center for *P. umbellatus*. This region contained the highest nucleotide diversity and unique genotypes (e.g., GB1, GB3). This conclusion is supported by accessions from Wenchuan, Sichuan (*n* = 3) and Luochuan, Shaanxi (*n* = 4), which exhibited three and four genotypes, respectively, and were distributed across three distinct clustering branches. The substantial genetic diversity detected suggests that this area serves as the primary genetic diversity center of the species. Notably, this finding is consistent with ecological evidence indicating that the Qinling Mountains, regarded as a major production region, contain the highest density of suitable habitats [59]. It also aligns with origin sites previously proposed based on ITS and LSU sequences (Shaanxi, Henan, and Gansu) [30], as these locations fall within the broader Qinling–Daba Mountain region.

The complexity of the population structure is further emphasized by the presence of Qinba-typical genetic components in the geographically distant Hengduan and Wumeng Mountain populations. Although this admixture suggests substantial gene flow, the pattern differs from a simple isolation-by-distance model. Considering the long history of *P. umbellatus* cultivation [26,60], historical human-mediated introduction of elite germplasm is a plausible explanation for this distribution. However, in the absence of explicit divergence-time estimates, these patterns should be interpreted primarily as evidence of a highly interconnected genetic network rather than definitive documentation of historical migration routes.

### 4.2. Validation of Six P. umbellatus Clades Through Extensive Sampling

Previous phylogenetic analyses of *P. umbellatus* were limited by sample size, hindering the resolution of geographic and phylogenetic relationships [27]. The present study employed an expanded dataset, the largest available so far, covering the species’ main habitats [59]. This comprehensive sampling allowed for the subdivision of *P. umbellatus* into six distinct clades, refining the four previously reported [28,30]. Importantly, the topology derived from ITS/LSU data strongly aligned with those obtained independently from genome-wide SNP and SSR markers. This agreement confirmed that the identified population structure is biologically accurate and representative across the species’ range.

Although rDNA markers are standard for fungal studies, reliance solely on these markers may cause issues due to intragenomic heterogeneity [61], potentially confounding species delineation and overestimating diversity [62,63]. Nevertheless, the consistency observed between nuclear ribosomal markers and genome-wide high-throughput sequencing (HTS) data in this study confirms the robustness of the reconstructed phylogeny. This cross-validation strongly indicates that the six-clade structure accurately reflects biological reality rather than methodological artifacts. Thus, multilocus or HTS approaches are essential for resolving complex fungal population structures [62].

### 4.3. Geographic Isolation Drives Genetic Divergence and Morphotype Differentiation

The identified genetic clades showed a significant positive correlation between genetic and geographic distances (Mantel test, *r* = 0.38, *p* < 0.001), confirming that isolation-by-distance (IBD) strongly contributes to divergence in *P. umbellatus*. This pattern mirrors observations in other fungi, where geographical barriers restrict gene flow [64,65,66]. Notably, the Northeast clade (Group 6) formed a separate group, supporting earlier findings [28,30,67]. Additionally, the Nujiang population in Yunnan (Group 3) displayed pronounced genetic uniqueness, likely due to rugged topography and habitat heterogeneity, creating effective barriers to gene exchange [68]. Although the sample size for this group is limited (*n* = 2), these accessions strictly satisfied the clustering criteria, suggesting a notably distinct lineage; however, we regard this as a preliminary observation requiring confirmation through broader sampling. Conversely, other clades exhibited genetic cohesion despite geographic separation. Examples include Group 1 (Qinba and Taihang Mountains) and Group 4 (Qilian and Hengduan Mountains). The clustering of Qilian and Hengduan Mountain populations suggests historical or geographical connections, though these relationships remain unclear and warrant additional investigation. However, the moderate Mantel correlation coefficient (*r* = 0.38) implies that geographic patterns have been partially disrupted. Such disruption may stem from high intra-population genetic diversity, illustrated by multiple genotypes within individual locations, such as Wangcang (Sichuan) and Lueyang (Shaanxi).

Importantly, these geographic and genetic patterns clarify the debated relationship between the two commercial morphotypes: Zhushiling and Jishiling. Although prior studies lacked sufficient samples to differentiate these clearly [30], this research revealed strong consistency between genetic clusters and morphotypes. Specifically, Groups 1, 3, and 4 primarily comprised the Zhushiling type, whereas Groups 5 and 6 predominantly contained the Jishiling type. The IBD pattern detected here enabled recognition of substantial genetic divergence among morphologically identical accessions from distinct mountain ranges. For instance, notable genetic differences were observed between Jishiling samples from Changbai and Wumeng Mountains and between Zhushiling from Qinba and Hengduan Mountains. These findings strongly indicate that these morphotypes are genetically determined rather than environmentally induced variants, providing a robust molecular foundation for the morphotaxonomy of *P. umbellatus*.

### 4.4. Genotype–Chemotype Coupling: Lineage-Specific Genetic Potential Amid Environmental Plasticity

Combined multivariate and univariate analyses established that genetic lineage primarily determines metabolite accumulation in *P. umbellatus*. PCA revealed distinct chemical signatures among genetic clusters, supported by significant statistical effect sizes. Specifically, Group 2 consistently exhibited the highest genetic potential for accumulating steroidal metabolites (polyporusterones A and B), whereas Group 4 had the highest average polysaccharide content (approximately two-fold higher than the lowest, Group 6). These findings highlight particular germplasms, Group 2 for steroids and Group 4 for polysaccharides, as promising targets for selective therapeutic breeding.

Nonetheless, interpreting these results requires consideration of the weak global correlation observed in the Mantel test (*r* = 0.09, *p* < 0.05). This low correlation underscores that while genetics provide the biological potential, the metabolic phenotype is heavily modulated by environmental plasticity. The “background noise” implies that factors such as soil edaphic properties, altitude, and microclimate play a substantial role in realizing these chemotypes. Thus, distinct lineage-specific profiles emerged despite this environmental variability, confirming that genetic regulation remains a robust, albeit not exclusive, determinant of quality.

### 4.5. Limited Predictive Value of Sclerotium Morphology and Future Perspectives

In contrast to the robust predictive power of genetic lineage (ϵ^2^ = 0.60), traditional morphotype classification (Zhushiling vs. Jishiling) demonstrated limited utility for assessing chemical quality. PCA based on morphotypes revealed substantial overlap, and while Zhushiling exhibited statistically higher polyporusterone A levels, the difference was moderate (1.28-fold, medium effect size *r* = 0.30). Furthermore, ergosterol concentrations remained consistent across all lineages and morphotypes (*p* = 0.804). Given its evolutionary conservation and critical membrane function [69], ergosterol serves as a reliable quality-control marker [4], but not as an indicator of superior germplasm. Regarding polysaccharides, the phenol-sulfuric acid method used here detects total carbohydrate biomass rather than specific bioactive fractions. Although morphotypes showed no significant quantitative differences (*p* > 0.05), this measure fails to reflect structural complexity relevant to bioactivity [1]. Research demonstrates that structural features profoundly influence the antitumor and immunomodulatory efficacy of *P. umbellatus* polysaccharides [70]. Thus, observed quantitative similarities do not exclude structural differences.

To close the quality loop, future research must move beyond simple quantification. First, studies should prioritize structural characterization (e.g., monosaccharide composition) and biological validation (e.g., cytotoxicity assays) to confirm that high-yield candidates like Group 4 possess superior pharmacological efficacy. Second, transcriptomic analyses (e.g., RNA-seq) are necessary to identify the upregulation of specific biosynthetic gene clusters in elite lineages, providing the molecular validation underpinning the observed chemotypes. Finally, to address the strong environmental influence, cultivation strategies must integrate Genotype—Environment interaction modeling. Correlating specific lineages with precise environmental data will allow for the development of comprehensive protocols, ensuring that farmers can reliably translate genetic potential into high-quality yields.

### 4.6. Genetic Erosion and Conservation Strategy: Balancing Inbreeding and Clonal Propagation

Population-level analysis revealed substantial heterozygote deficits across all genetic groups (*H*_o_ = 0.18 vs. *H*_e_ = 0.48), significantly deviating from Hardy–Weinberg equilibrium (*F*_IS_ = 0.61). This excess homozygosity presents a paradox, given the heterothallic, tetrapolar mating system of *P. umbellatus*, which theoretically promotes outcrossing [2,71]. However, local ecological conditions and human-induced pressures might override this biological system. Habitat fragmentation restricts spore dispersal [72], forcing sibling mating among neighboring monokaryotic mycelia [64]. This scenario is indirectly supported by pronounced genetic differentiation observed among geographically distant populations, indicating restricted gene flow—a macro-level consequence of localized inbreeding. Moreover, the prevalent commercial practice of long-term clonal propagation via sclerotial fragmentation exacerbates this forced inbreeding [2]. Without sexual recombination, clonal lines become genetically uniform, hindering the removal of deleterious alleles [73]. The resulting decline in genetic diversity negatively impacts individual fitness, population dynamics, and long-term evolutionary potential [74,75], making populations vulnerable to genetic drift and further diversity loss [58].

To mitigate genetic erosion, a two-pronged conservation strategy is necessary. In situ conservation should prioritize habitat restoration to reconnect fragmented populations and encourage natural gene flow. Ex situ breeding must transition from clonal propagation to sexual recombination, generating new allele combinations and enhancing progeny adaptability [76]. Recent advances by Li et al., demonstrating viable hybridization between monokaryotic conidial and basidiospore isolates [77], provide essential technical groundwork for genetic improvement. Leveraging this breakthrough, our study establishes a basis for selecting superior germplasms—specifically, Group 2 for steroids and Group 4 for polysaccharides. Nevertheless, practical implementation faces challenges. Despite the feasibility of laboratory hybridization, the lengthy vegetative growth period (3–4 years) [78] delays screening of superior hybrids. Furthermore, regulatory protocols must evolve to recognize molecularly distinct cultivars. Overcoming these biological and regulatory barriers is essential for sustainable modernization of the *P. umbellatus* industry.

## 5. Conclusions

This study establishes the first high-resolution genetic–geographic–chemical framework for *Polyporus umbellatus* by combining whole-genome resequencing and metabolic profiling. Six distinct genetic clusters were identified nationwide, with the Qinling–Daba Mountains as the putative diversity center. Crucially, genetic lineage, not traditional morphotype classification (Zhushiling and Jishiling), primarily determines chemical variation. Specifically, Group 2 (consistent steroid accumulation) and Group 4 (high polysaccharide yield) represent superior germplasm candidates. Additionally, pronounced heterozygote deficits indicate severe genetic erosion from inbreeding and prolonged clonal propagation, highlighting the urgent need for in situ restoration and adoption of sexual breeding. Regarding industrial applications, a paradigm shift in quality control is recommended: ergosterol remains a stable biomass marker, but assessments should transition from unreliable morphological criteria to molecular-assisted selection and structural polysaccharide characterization. This approach will provide a scientifically rigorous foundation to update pharmacopeial standards and advance precision breeding.

## Figures and Tables

**Figure 1 jof-12-00039-f001:**
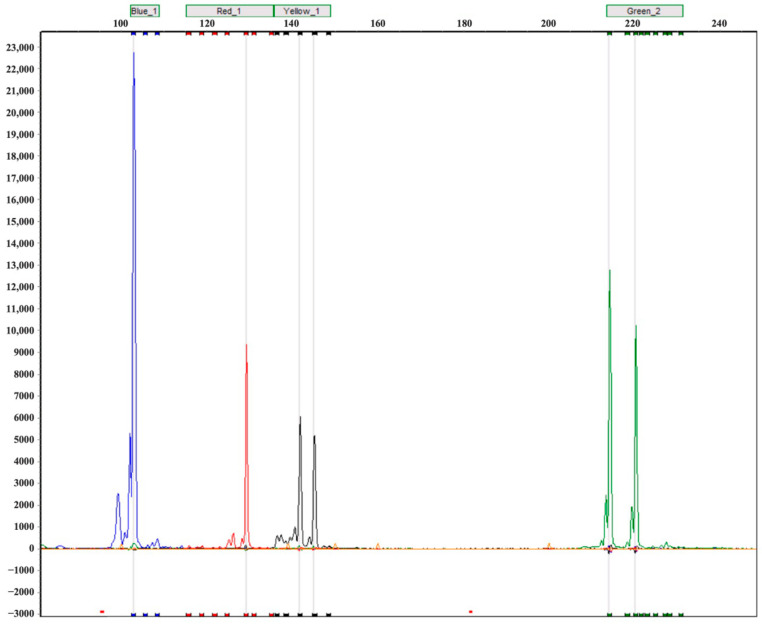
Allelic variations in *P. umbellatus* samples amplified using four SSR primers (Blue_1, Red_1, Yellow_1 [visualized as black peaks], and Green_2), visualized by fluorescent capillary electrophoresis. The vertical axis represents fluorescence intensity, while the horizontal axis indicates allele size (bp). A single peak signifies homozygosity; two distinct peaks represent heterozygosity.

**Figure 2 jof-12-00039-f002:**
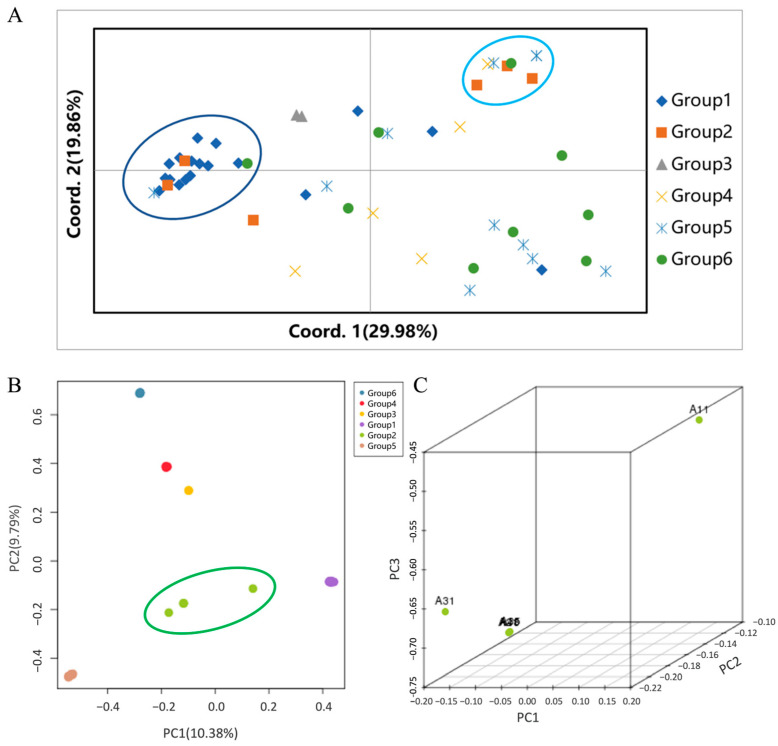
Ordination analyses illustrating genetic diversity in *P. umbellatus* populations. (**A**) PCoA based on SSR markers; blue ellipses highlight the distinct Group 1 and the overlapping Groups 2–6. (**B**) Global PCA based on SNP markers (2D view) the green ellipse highlights the considerable scatter of Group 2, indicating higher internal genetic divergence. (**C**) 3D PCA highlighting internal genetic substructure within Group 2.

**Figure 3 jof-12-00039-f003:**
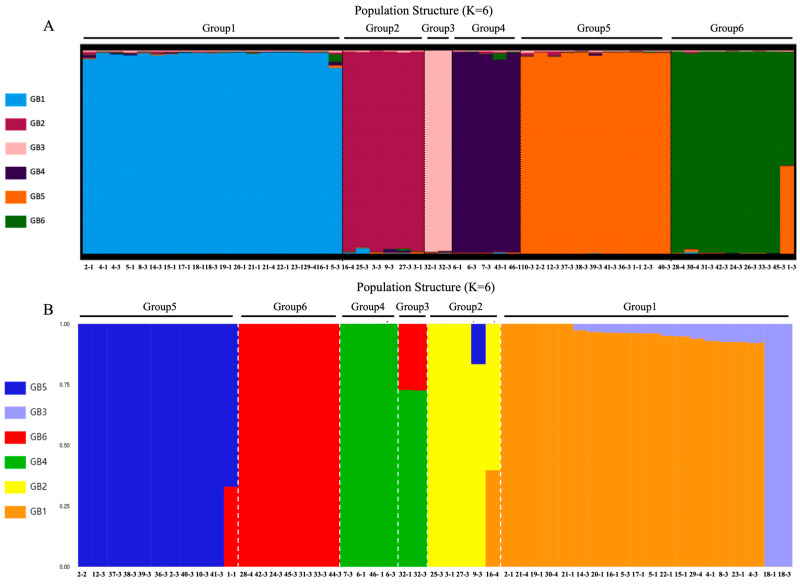
Population genetic structure of *P. umbellatus*. (**A**) Bayesian clustering of 53 germplasms based on 19 core SSR markers (*K* = 6) using STRUCTURE v2.3.4. (**B**) Maximum likelihood clustering of 47 accessions using SNP data (*K* = 6) via ADMIXTURE v1.22. Each vertical bar represents one individual, with colors indicating proportional membership (*Q*) in the six inferred genotype (GB1–GB6).

**Figure 4 jof-12-00039-f004:**
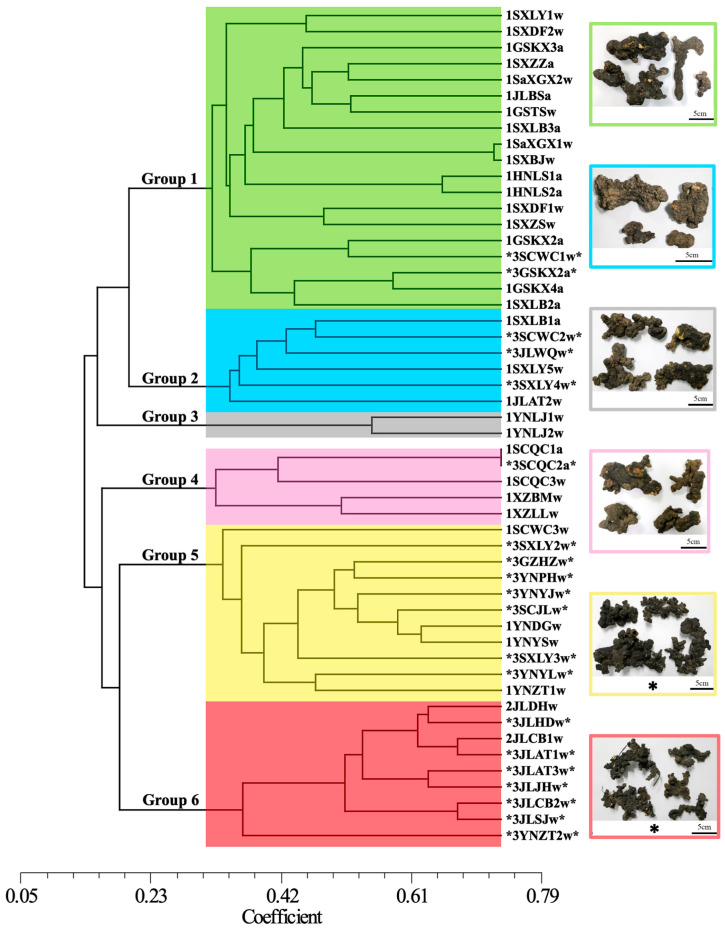
UPGMA dendrogram of 53 *P. umbellatus* germplasms. The dendrogram illustrates six genetic groups indicated by colored branches: Group 1 (green), Group 2 (blue), Group 3 (grey), Group 4 (pink), Group 5 (yellow), and Group 6 (red). Asterisks (*) denote germplasms of the Jishiling type. Representative sclerotium morphologies for each group appear on the right, with border colors matching their respective groups.

**Figure 5 jof-12-00039-f005:**
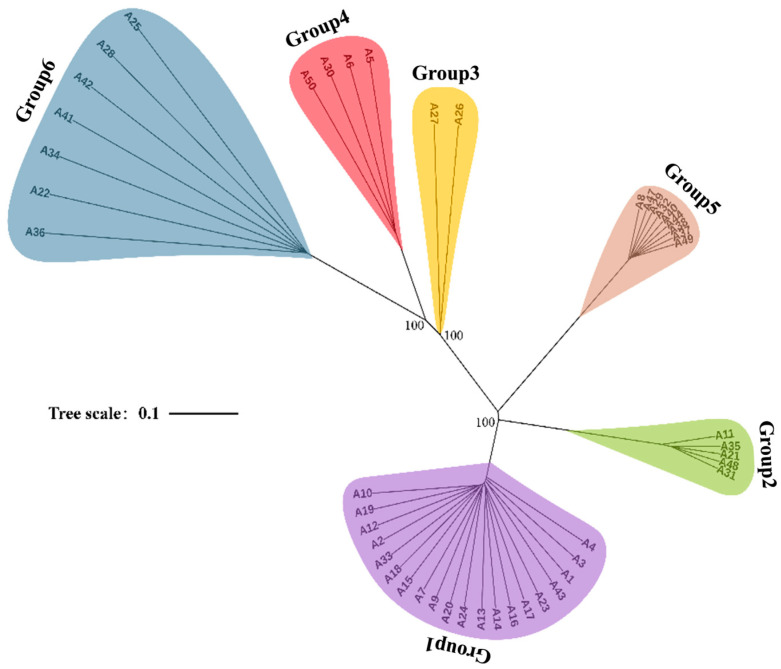
Maximum likelihood (ML) phylogenetic tree of 47 accessions based on genomic SNPs. The tree confirms the six genetic lineages (Groups 1–6) identified by UPGMA clustering, with different background colors distinguishing each group. Bootstrap support values (100%) at key nodes indicate high confidence in the separation of these groups.

**Figure 6 jof-12-00039-f006:**
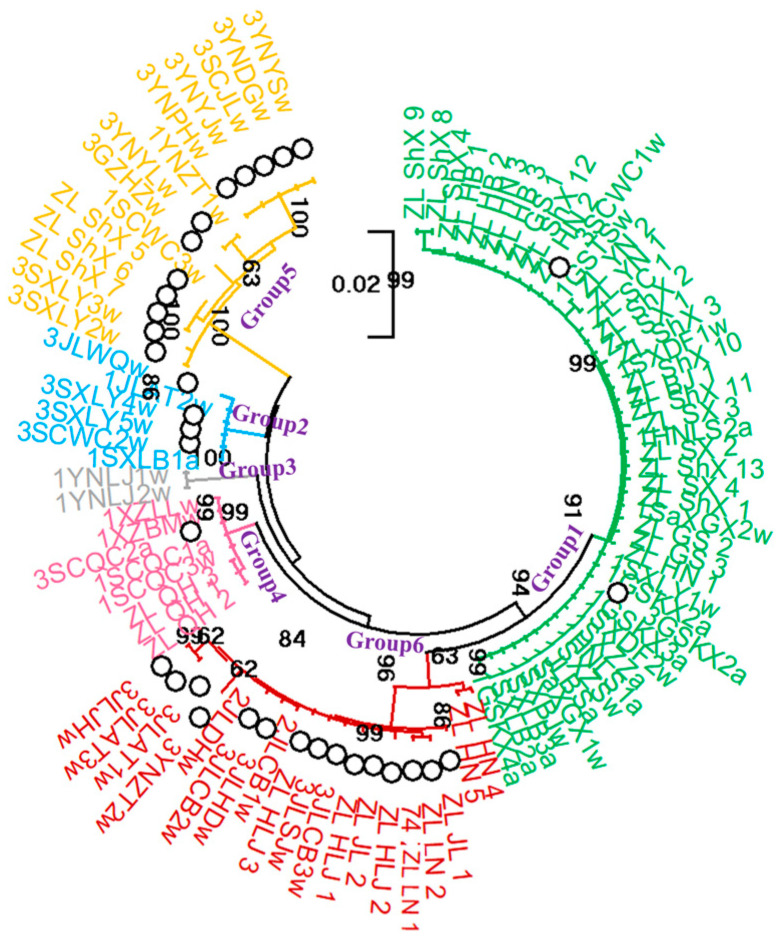
Maximum likelihood (ML) phylogenetic tree of 95 *P. umbellatus* germplasms based on concatenated ITS and LSU sequences. The different colors of the branches and labels indicate the six identified groups (Groups 1–6). Circles (○) indicate Jishiling samples.

**Figure 7 jof-12-00039-f007:**
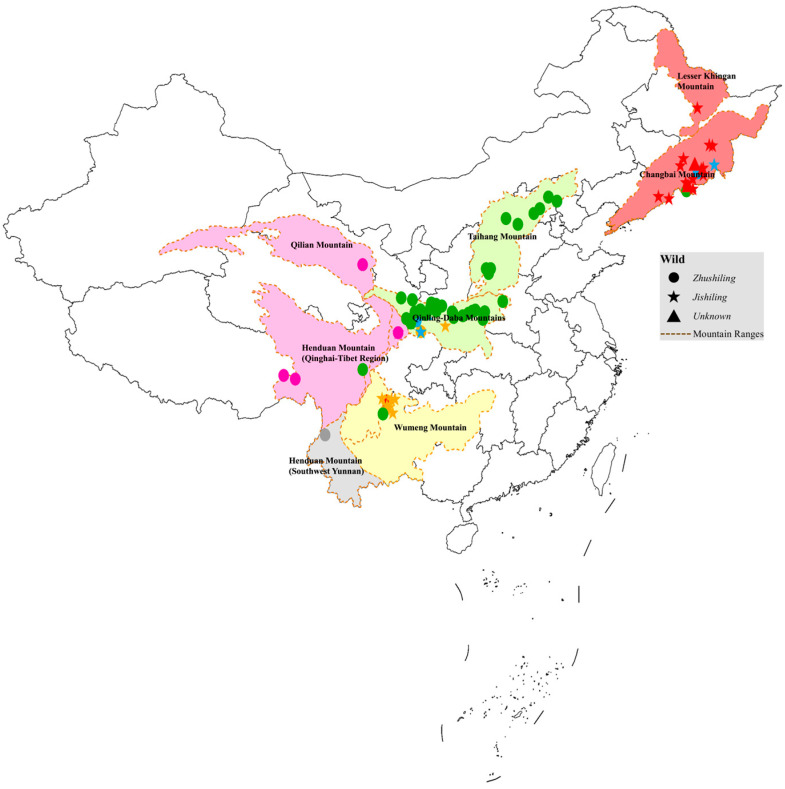
Genetic clustering of 95 *P. umbellatus* germplasms integrated with a geographical distribution map. The different colored circles and stars represent the genetic groups identified in the ITS phylogenetic tree (Figure 6), with green indicating Group 1, blue indicating Group 2, grey indicating Group 3, pink indicating Group 4, yellow indicating Group 5, and red indicating Group 6.

**Figure 8 jof-12-00039-f008:**
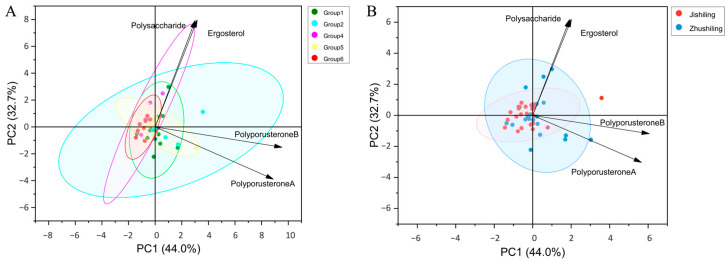
PCA biplots of four medicinal components in *P. umbellatus* (76.7% total variance). Vectors indicate component loadings; ellipses represent 95% confidence intervals. (**A**) Samples color-coded by genetic group show distinct clusters, notably Groups 2 and 4. (**B**) Samples colored by morphotype, demonstrating substantial overlap.

**Figure 9 jof-12-00039-f009:**
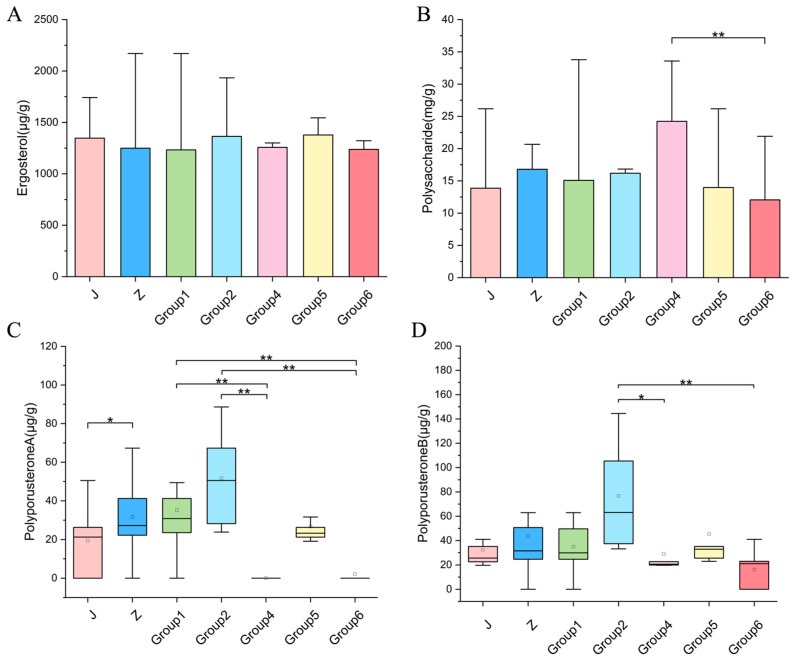
Comparison of four medicinal components among Jishiling (J), Zhushiling (Z), and genetic groups. (**A**) Ergosterol; (**B**) Polysaccharide; (**C**) Polyporusterone A; (**D**) Polyporusterone B. The small squares inside the box plots represent the mean values. (* *p* < 0.05; ** *p* < 0.01).

**Table 1 jof-12-00039-t001:** Primer sequences and characteristics of 20 microsatellite markers developed for *P. umbellatus*.

No.	Primer	Type	Direction	Sequence (5′–3′)	Repeat Motif	T_A_ (°C)	Allelic Size
A	F1	gSSR	F	CCCCGCGAAGACTTGTATGA	(AT)_15_	56	232
			R	TGTATGTGCTTACTGGCCCG			
O	F15	gSSR	F	CGACTACGGTCAGCCACTAC	(AG)_12_	58	151
			R	TCTCTCTCTCTGCGTCGTCA			
P	F16	gSSR	F	ACACGTGTGGGATTCAACGT	(AC)_11_	54	230
			R	ACGCATCCATGAACGTCTGA			
R	F18	gSSR	F	GCATTGTGTTGGGCCAAGAG	(AG)_8_	56	111
			R	ACCAGAACCTTCCTTGTGCC			
S	F19	gSSR	F	GTAGACCGTGCTAGTGGCTC	(TGTA)_6_	58	144
			R	GGGTGAGAGTGTGACTTGGG			
U	F21	gSSR	F	TCGTTGACCTTGCCTTGTCT	(GTAA)_5_	54	109
			R	GCGTCAGATTGACCACTCCA			
W	F23	gSSR	F	CATCCCCCTCGCATGATACC	(TGC)_8_	58	207
			R	GCGGGGTACTATTTGCCGTA			
Z	F26	gSSR	F	GTTGTGCTGCTGGGCTATTG	(ATT)_6_	56	221
			R	CAAGCCCTGCTGTGAAAACC			
f	F32	gSSR	F	TGTGCCTTTCGCCATCTCTT	(TC)_9_	54	109
			R	GAGAGGGAGGCTACTGACCA			
r	F44	gSSR	F	GGACCACCGCCAATTTGTTC	(TCA)_8_	56	148
			R	GGGTTCCCAGTCACAGGATG			
s	F45	gSSR	F	GGTGTACGAGGTGGAAGCAA	(GGC)_8_	56	103
			R	CGCGAAGAAAGCCCAGAATG			
t	F46	gSSR	F	TGGTGTGCCCAACTTTAGCA	(GAT)_8_	54	118
			R	GTGTCCGATCACTAGCCTCG			
F	F6	gSSR	F	TGGGAATGGGTAGTCCGAGT	(GGA)_13_	56	112
			R	AATGAGCGTCGTCATTTGCG			
H	F8	gSSR	F	TTGTTGGCGGTTGCAATCAG	(CCT)_12_	54	131
			R	TTGGTTCGTAGGACGTGGTG			
β	PUF14	EST-SSR	F	CTCGCATCTCCACCATCTCC	(CGC)_5_	58	250
			R	CCTCTCACTTTCCCTCGAGC			
ε	PUF16	EST-SSR	F	GTCCCTGTAGTCGCTTCTCG	(GAC)_7_	56	230
			R	AGTTGGAGAGACAAGCGTGG			
ζ	PUF17	EST-SSR	F	CCAGACATGCTCGACACTCA	(TCCA)_5_	56	130
			R	GTGATGGATGTGGGGAAGGG			
η	PUF31	EST-SSR	F	CCAAGACCCCGCAAACCTAT	(GGC)_5_	56	140
			R	GTTGGGTGTGGCGAATTTCC			
γ	PUF33	EST-SSR	F	ATCCTCAGAGTCACCCCCTC	(TCA)_6_	58	270
			R	CGACGCGAGGATGAGAATGA			
δ	PUF35	EST-SSR	F	CTTTCTTGCGTGCCCTTTCC	(GCT)_6_	58	300
			R	AAGGTCAGGAATGCTTCGGG			

Note: F, Forward primer; R, Reverse primer.

**Table 2 jof-12-00039-t002:** Genetic parameters for the 20 SSR markers.

Primers	No. of Observed Allele	No. of Effective Alleles	Shannon’s Information Index	Observed Heterozygosity	Expected Heterozygosity	Polymorphic Information Content
	(*N*_a_)	(*N*_e_)	(*I*)	(*H*_o_)	(*H*_e_)	(PIC)
F15	2	2	0.69	0.62	0.5	0.37
F16	10	4.88	1.79	0.31	0.8	0.77
F19	2	1.66	0.59	0.1	0.4	0.32
F21	2	1.27	0.37	0.11	0.21	0.19
F26	6	3.79	1.5	0.11	0.74	0.69
F44	5	2.3	1.13	0.38	0.56	0.53
F45	3	2.81	1.06	0.2	0.64	0.57
F46	7	5.16	1.76	0.35	0.81	0.78
F6	3	1.87	0.72	0.1	0.46	0.37
PUF14	5	3.27	1.31	0.34	0.69	0.64
PUF16	4	3.35	1.3	0.21	0.7	0.65
PUF17	3	1.65	0.69	0	0.39	0.35
PUF31	4	1.92	0.92	0.05	0.48	0.44
PUF33	7	3.08	1.39	0.06	0.68	0.64
PUF35	9	4.85	1.83	0.37	0.79	0.77
Mean	5	2.92	1.14	0.22	0.59	0.54

**Table 3 jof-12-00039-t003:** AMOVA results for *P. umbellatus* populations.

Source of Variation	Degrees of Freedom (*df*)	Sum of Squares (SS)	Mean Square (MS)	Estimated Variance (Est. Var.)	Percentage
Among Populations	5	101.139	20.228	0.803	15%
Among Individuals	46	340.342	7.399	2.964	57%
Within Individuals	52	76.500	1.471	1.471	28%
Total	103	517.981	29.098	5.238	100%

**Table 4 jof-12-00039-t004:** Genetic diversity parameters among different genetic clusters.

	*N* _a_	*N* _e_	*I*	*H* _o_	*H* _e_	PIC	*F* _IS_
Group1	4	2.28	0.86	0.32	0.46	0.28	0.30
Group2	3	2.14	0.77	0.19	0.45	0.22	0.58
Group4	2	2.17	0.78	0.09	0.50	0.13	0.82
Group5	3	2.17	0.84	0.15	0.50	0.14	0.70
Group6	3	2.06	0.81	0.17	0.47	0.10	0.64
Mean	3	2.16	0.81	0.18	0.48	0.17	0.61

**Table 5 jof-12-00039-t005:** Genetic diversity parameters of different mountain populations.

	*N* _a_	*N* _e_	*I*	*H* _o_	*H* _e_	PIC	*F* _IS_
Qinba Mountain	4	2.58	0.98	0.29	0.51	0.41	0.43
Hengduan Mountain	3	2.61	0.96	0.14	0.55	0.28	0.75
Changbai Mountain	3	2.16	0.86	0.18	0.51	0.32	0.65
Wumeng Mountain	3	2.08	0.80	0.14	0.49	0.17	0.71
Mean	3	2.35	0.90	0.19	0.52	0.30	0.63

**Table 6 jof-12-00039-t006:** Genetic diversity parameters of Zhushiling and Jishiling.

	*N* _a_	*N* _e_	*I*	*H* _o_	*H* _e_	PIC	*F* _IS_
Zhushiling	5	2.77	1.07	0.27	0.54	0.46	0.50
Jishiling	3	2.28	0.91	0.15	0.53	0.43	0.72
Mean	4	2.52	0.99	0.21	0.54	0.45	0.61

## Data Availability

The original contributions presented in this study are included in the article/Appendix A. Further inquiries can be directed to the corresponding authors.

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
