# Peer review of "Genetic–Geographic–Chemical Framework of *Polyporus umbellatus* Reveals Lineage-Specific Chemotypes for Elite Medicinal Line Breeding"

_jof, 2026, doi:10.3390/jof12010039_

Round 1

Reviewer 1 Report (Previous Reviewer 3)

The authors carefully revised point-by-point my suggestions (methods, discussion, conclusions, formatting) and they provided an appropriated version for publication in JoF.

The authors carefully revised point-by-point my suggestions and they provided an appropriated version for publication in JoF.

Author Response

We are grateful for the reviewer's time and constructive feedback throughout this process. We are delighted that the revised version is now considered appropriate for publication.

Reviewer 2 Report (Previous Reviewer 4)

The manuscript was sufficiently improved and could be published. 

See above

Author Response

We sincerely thank the reviewer for their positive assessment and recommendation for publication.

Reviewer 3 Report (New Reviewer)

In this article, self-citation was also detected, although in my view, self-citation is mainly used to establish the genomic framework of the study. In references [2], [25], and [58],

The article should address certain questions that lie between the DNA sequence and the final pharmaceutical product.

Genetic-Environment Interaction (G×E) Modeling

The authors conclude that the correlation between genetic groups and chemical compounds is "weak" overall. This suggests that the environment (microclimate, soil type, host tree) has a huge impact on the quality of the fungus.

What is suggested: An analysis of environmental variables (soil science, altitude, precipitation) of the sampling sites. Without this, a farmer might buy the "Elite Lineage" (Group 2) but not get the same steroids if the soil on their farm is different.

Biological Validation of Chemotypes (Expression Studies)

The study identifies which lineages have more chemicals, but it does not explain why at the biological level.

What is suggested: Transcriptomic analysis (RNA-seq). It would be much more robust if they could identify which specific genes in the steroid biosynthetic pathway are being "turned on" in Group 2. This would confirm that the superiority is a fixed genetic trait and not a temporary response to environmental stress.

Greater Representation of Minority Groups

Statistical rigor weakens in smaller groups.

What is suggested: Increase the sample size of Group 3 (n=2). In population statistics, two individuals are not enough to define an entire lineage with certainty. For the proposed framework to be universal, an equitable representation of all the mountainous regions of China is needed.

Biological Activity Tests (Bioassays) 

The article assumes that "more chemicals = better medicine," which is a chemical logic but not necessarily a pharmacological one. 

What is suggested: Cytotoxicity or immunological activity assays in vitro. Demonstrating that extracts from Group 4 (rich in polysaccharides) actually inhibit tumor cells more effectively than the other groups would definitively close the quality loop. 

On the other hand, the article features a highly sophisticated and appropriate statistical methodology. In this case, it is because the relationship between three dimensions was validated: genetics, geography, and chemistry.

In this article, self-citation was also detected, although in my view, self-citation is mainly used to establish the genomic framework of the study. In references [2], [25], and [58],

The article should address certain questions that lie between the DNA sequence and the final pharmaceutical product.

Genetic-Environment Interaction (G×E) Modeling

The authors conclude that the correlation between genetic groups and chemical compounds is "weak" overall. This suggests that the environment (microclimate, soil type, host tree) has a huge impact on the quality of the fungus.

What is suggested: An analysis of environmental variables (soil science, altitude, precipitation) of the sampling sites. Without this, a farmer might buy the "Elite Lineage" (Group 2) but not get the same steroids if the soil on their farm is different.

Biological Validation of Chemotypes (Expression Studies)

The study identifies which lineages have more chemicals, but it does not explain why at the biological level.

What is suggested: Transcriptomic analysis (RNA-seq). It would be much more robust if they could identify which specific genes in the steroid biosynthetic pathway are being "turned on" in Group 2. This would confirm that the superiority is a fixed genetic trait and not a temporary response to environmental stress.

Greater Representation of Minority Groups

Statistical rigor weakens in smaller groups.

What is suggested: Increase the sample size of Group 3 (n=2). In population statistics, two individuals are not enough to define an entire lineage with certainty. For the proposed framework to be universal, an equitable representation of all the mountainous regions of China is needed.

Biological Activity Tests (Bioassays) 

The article assumes that "more chemicals = better medicine," which is a chemical logic but not necessarily a pharmacological one. 

What is suggested: Cytotoxicity or immunological activity assays in vitro. Demonstrating that extracts from Group 4 (rich in polysaccharides) actually inhibit tumor cells more effectively than the other groups would definitively close the quality loop. 

On the other hand, the article features a highly sophisticated and appropriate statistical methodology. In this case, it is because the relationship between three dimensions was validated: genetics, geography, and chemistry.

Author Response

1. Summary

We sincerely thank Reviewer 3 for the thorough review and highly constructive comments. The reviewer has identified several crucial areas—specifically regarding environmental variables, transcriptomic validation, and bioactivity assays—that represent valuable entry points for in-depth research. While we have carefully considered these suggestions, conducting additional wet-lab experiments is not feasible at this stage due to the long growth cycle of P. umbellatus and current logistical constraints.

However, we believe that identifying these interesting phenomena and research gaps is exactly where the value of this study lies. Therefore, rather than adding new data, we have significantly revised the Discussion section (specifically Sections 4.4 and 4.5) to incorporate these points. By doing so, we aim to provide clear, strategic directions for readers and our research team to conduct these necessary follow-up studies.

2. Point-by-point response to Comments and Suggestions for Authors

Comment 1: In this article, self-citation was also detected, although in my view, self-citation is mainly used to establish the genomic framework of the study. In references [2], [25], and [58],

Response 1: We appreciate the reviewer’s careful attention to our references. We respectfully clarify that References [25] and [58] are not self-citations, as they originate from independent research groups.

Regarding the actual self-citations, we appreciate the reviewer acknowledging that they are used to establish the study's framework. Since Polyporus umbellatus is a specialized research field, our previous works provide the necessary foundation for the current study:

•      Ref [2] reports the first genome assembly of P. umbellatus, which, as the reviewer noted, establishes the essential genomic framework for this research.

•      Ref [32] is cited to introduce the specific research progress on the two sclerotial forms (Zhushiling and Jishiling), which is critical context for our comparative analysis.

•      Ref [77] demonstrates viable hybridization techniques, providing the essential technical groundwork for the genetic improvement discussion in our manuscript.

We have ensured that all citations are strictly relevant and necessary to support the scientific logic of the article.

Comment 2: The article should address certain questions that lie between the DNA sequence and the final pharmaceutical product.

Response 2: We appreciate this insightful comment. We recognize that bridging the gap between "genotype" and the "final pharmaceutical product" requires addressing gene expression and biological activity. In the revised Section 4.5, we explicitly proposed transcriptomic analyses (RNA-seq) to validate the molecular mechanisms and bioassays (e.g., cytotoxicity tests) to confirm pharmacological efficacy. Furthermore, we integrated Genotype - Environment modeling to ensure that this genetic potential can be reliably translated into the final medicinal product.

Comment 3: Genetic-Environment Interaction (G×E) Modeling

The authors conclude that the correlation between genetic groups and chemical compounds is "weak" overall. This suggests that the environment (microclimate, soil type, host tree) has a huge impact on the quality of the fungus.

What is suggested: An analysis of environmental variables (soil science, altitude, precipitation) of the sampling sites. Without this, a farmer might buy the "Elite Lineage" (Group 2) but not get the same steroids if the soil on their farm is different.

Response 3: We fully agree. We have revised Section 4.4 to acknowledge that the weak Mantel correlation implies the metabolic phenotype is heavily modulated by environmental plasticity. Furthermore, in Section 4.5, we explicitly proposed that future cultivation strategies must integrate " Genetic-Environment interaction modeling." We emphasized that correlating specific lineages with precise environmental data is essential to ensure farmers can reliably translate genetic potential into high-quality yields.

Comment 4: Biological Validation of Chemotypes (Expression Studies)

The study identifies which lineages have more chemicals, but it does not explain why at the biological level.

What is suggested: Transcriptomic analysis (RNA-seq). It would be much more robust if they could identify which specific genes in the steroid biosynthetic pathway are being "turned on" in Group 2. This would confirm that the superiority is a fixed genetic trait and not a temporary response to environmental stress.

Response 4: Agreed. To address the molecular mechanism, we updated Section 4.5 to emphasize that "transcriptomic analyses (e.g., RNA-seq) are necessary." We explicitly state that identifying the upregulation of specific biosynthetic gene clusters is required to confirm that the superiority of Group 2 is a fixed genetic trait rather than a temporary response to environmental stress.

Comment 5: Greater Representation of Minority Groups

Statistical rigor weakens in smaller groups.

What is suggested: Increase the sample size of Group 3 (n=2). In population statistics, two individuals are not enough to define an entire lineage with certainty. For the proposed framework to be universal, an equitable representation of all the mountainous regions of China is needed.

Response 5: We acknowledge the limited sample size for the Nujiang population, attributed to the logistical challenges of locating subterranean sclerotia in such rugged topography. In the revised Section 4.3, we explicitly designated this finding as a "preliminary observation" pending broader sampling. However, we emphasized that these accessions strictly satisfied clustering criteria, strongly indicating a distinct endemic lineage despite the small number.

Comment 6: Biological Activity Tests (Bioassays) 

The article assumes that "more chemicals = better medicine," which is a chemical logic but not necessarily a pharmacological one. 

What is suggested: Cytotoxicity or immunological activity assays in vitro. Demonstrating that extracts from Group 4 (rich in polysaccharides) actually inhibit tumor cells more effectively than the other groups would definitively close the quality loop. 

Response 6: We agree that chemical yield does not strictly equate to clinical efficacy. In the revised Section 4.5, we clarified this distinction and added that future research must prioritize "biological validation (e.g., cytotoxicity assays)." This addition confirms that bridging the gap between chemotype (high content) and phenotype (high activity) is the critical final step to close the quality loop for P. umbellatus breeding.

Comment 7: On the other hand, the article features a highly sophisticated and appropriate statistical methodology. In this case, it is because the relationship between three dimensions was validated: genetics, geography, and chemistry.

Response 7: We appreciate the reviewer’s encouraging comments regarding our methodology. The integration of genetics, geography, and chemistry was indeed critical for establishing the reliability of our findings.

This manuscript is a resubmission of an earlier submission. The following is a list of the peer review reports and author responses from that submission.

Round 1

Reviewer 1 Report

The manuscript should be rejected and resubmitted after revision. The reasons for this decision are: 1) It is too long, and authors should make it concise explaining the important and novel aspects of the work. 2) many figures are inserted twice, and some are without figure captions which is confusing. 3) There are several formatting issues in the manuscript.

Several figures are inserted twice in the manuscript and should be fixed.

Author Response

1. Summary

Thank you very much for taking the time to review our manuscript. We genuinely appreciate your feedback regarding the manuscript's length and formatting. We have carefully addressed your concerns by streamlining the text, removing redundant figures, and correcting formatting errors. Detailed responses are listed below, and all changes have been highlighted in the re-submitted files.

2. Point-by-point response to Comments and Suggestions for Authors

Comment 1: It is too long, and authors should make it concise explaining the important and novel aspects of the work.

Response 1: We gratefully acknowledge this suggestion. We have extensively streamlined the manuscript to improve conciseness. Specifically, we shortened the Abstract and Discussion by removing redundancy and excessive details, thereby sharpening the focus on the novel "genetics–geography–chemistry" framework and the discovery of lineage-specific quality determinants.

Comment 2: Many figures are inserted twice, and some are without figure captions which is confusing.

Response 2: We sincerely apologize for this oversight. We have carefully checked the file, removed all duplicate figures, and ensured that every figure is correctly positioned with its corresponding caption.

Comment 3: There are several formatting issues in the manuscript.

Response 3: We apologize for these errors. We have thoroughly proofread the manuscript to correct all formatting inconsistencies and ensured strict adherence to the journal’s guidelines.

3. Response to Comments on the Quality of English Language

Response: The manuscript has been professionally edited by a native English-speaking team to ensure the language meets academic standards.

Reviewer 2 Report

Dear Authors,

I recommend major revision, and I have included all of my suggestions in the attached reviewer report.

Review report

Journal: Journal of Fungi

Manuscript ID: jof-3972349

Dear authors,

Please find all my suggestions below:

Abstract (page 1, lines 10-35): I suggest shortening the Abstract, as it is currently too long in my opinion. The key findings and conclusions should be presented more concisely to improve clarity and readability. I suggest to you revise:

  • Overly long sentences; reduce complexity.
  • Remove excessive numerical detail.
  • Add 1–2 sentences explaining significance for breeding and conservation

Keywords

  • Replace “chemical component” with “metabolite profiling” or “chemotype.”

  1. Introduction (pages 2-3, lines 41-105)

The Introduction is nearly 2 pages and contains excessive background (history of pharmacopoeia, long lists of bioactivities, and details on cultivation history).
The narrative sometimes drifts into a mini-review. I want to recommend to you focus on key points:

  • Why P. umbellatus is important
  • Why genetic and chemical variability is problematic
  • Knowledge gaps
  • Clear objective statement that could be done by revision of the current objectives

Specific issues:

  • The listing of bioactivities (lines 51–62) is too long and can be shortened and revised.
  • Paragraphs discussing legal protection (lines 63–74) are too detailed for introduction.
  • Repetition of “heterogeneous germplasm” and “high variation” occurs multiple times – please revise this.

  1. Materials and Methods (pages 3-6, lines 107-245):

I suggest that you provide clearer structuring and remove redundant details in this segment of your research article.

The Methods section reads as a long technical protocol, especially SSR and SNP workflows.
Some parts appear copied from standard lab manuals, please revise this part of Materials and methods.

Generally my suggestions include the following recommendations:

  • Summarize PCR conditions and reference established protocols rather than describing every step in detail.
  • Move lengthy primer design parameters (Primer3 settings, PCR volumes, PAGE descriptions) to Supplementary Material.
  • Clarify why only 47 of 53 samples had sclerotia and had metabolite profiling, and how this uneven sampling may influence chemotype interpretation.

  1. Results (pages 6-24, lines 246-515)

You included excessive figures and some of them need consolidation.

In general, there are too many figures for the same result type (multiple PCA plots, multiple structure plots).

To be revised:

  • PCA figures for SSR and SNP appear redundant.
  • STRUCTURE and ADMIXTURE outputs could be merged.
  • Some dendrograms are difficult to read due to small fonts, especially colored branches.

In general, I want to recommend following:
Combine related figures:

  • Merge PCA results into one panel (SSR + SNP).
  • Merge STRUCTURE and ADMIXTURE outputs for both marker types.
  • Improve font size, resize labels, and simplify color schemes.

Regarding statistical analysis of chemical data requires clarification, because only four compounds were measured; interpretation of “chemotype clusters” may be overstated.

Recommendation:
You could include:

  • Normality tests (Shapiro–Wilk or Kolmogorov–Smirnov)
  • Post hoc tests (Dunn’s or Tukey)
  • Effect sizes (η² or r)
  • A PCA of chemical data could provide stronger chemotype separation.

  1. Discussion (pages 24-28, lines 516-711)

This segment is too long and partially repetitive. It restates the results across multiple paragraphs, which affects clarity and readability. I recommend shortening and streamlining it to avoid redundancy. Also, avoid citing too many references when not necessary (e.g., rDNA evolution discussion), and please clearly separate “genotype–chemotype relations” from “geographic structure.”

I suggest to you:

  • Remove sections that repeat the Results almost verbatim.
  • Reduce speculative statements regarding origin, historical dispersal, and domestication unless literature supports these claims.
  • Shorten section 4.1 and 4.2, which re-explain SSR/SNP methods.

  1. Conclusions (pages 28-29, lines 712-728)

The paper ends abruptly without a structured conclusion. Could you please add a concise Conclusions summarizing six clades established, Qinling–Daba as putative diversity center, Clear genotype–chemotype link, and at the end implications for breeding, conservation, and pharmacopoeial standardization

Minor comments regarding aubmitted Figures:

  • UPGMA dendrogram: labels too small; colors too similar.
  • STRUCTURE barplot: consider grouping by geographic region.
  • ML tree: colors are good, but bootstrap values should be more visible.

I recommend Major revision.

My general impression is that your manuscript is valuable and likely publishable after substantial revision.

Key priorities include:

  1. Condensing the Introduction and Discussion.
  2. Streamlining the Methods and moving technical details to the Supplementary Materials.
  3. Integrating the SSR and SNP results more clearly.
  4. Improving the quality and clarity of the figures.
  5. Toning down speculative or unsupported conclusions.

Author Response

1. Summary

Thank you very much for your thorough review and constructive comments. We genuinely appreciate your detailed suggestions regarding the manuscript's structure, statistical analysis, and figure presentation. We have carefully addressed each point, particularly focusing on streamlining the Introduction and Discussion, refining the statistical methods for chemical analysis, and consolidating the figures.

Please find our detailed point-by-point responses below. All corresponding revisions have been highlighted in the re-submitted files.

2. Point-by-point response to Comments and Suggestions for Authors

Comments 1: Abstract (page 1, lines 10-35): 

I suggest shortening the Abstract, as it is currently too long in my opinion. The key findings and conclusions should be presented more concisely to improve clarity and readability. I suggest to you revise:

  • Overly long sentences; reduce complexity.
  • Remove excessive numerical detail.
  • Add 1–2 sentences explaining significance for breeding and conservation

Response 1: We have thoroughly revised the Abstract to enhance clarity and readability. Specifically, we simplified complex sentence structures and removed excessive numerical details (e.g., specific SNP counts and raw test statistics). We also added a concluding statement explicitly highlighting the study's practical implications for urgent in situ conservation and precision breeding strategies.

Comments 2: Keywords

  • Replace “chemical component” with “metabolite profiling” or “chemotype.”

Response 2: Agreed. We have replaced this term with 'metabolite profiling' as suggested.

Comments 3: Introduction (pages 2-3, lines 41-105)

The Introduction is nearly 2 pages and contains excessive background (history of pharmacopoeia, long lists of bioactivities, and details on cultivation history).
The narrative sometimes drifts into a mini-review. I want to recommend to you focus on key points:

  • Why P. umbellatus is important
  • Why genetic and chemical variability is problematic
  • Knowledge gaps
  • Clear objective statement that could be done by revision of the current objectives

Specific issues:

  • The listing of bioactivities (lines 51–62) is too long and can be shortened and revised.
  • Paragraphs discussing legal protection (lines 63–74) are too detailed for introduction.
  • Repetition of “heterogeneous germplasm” and “high variation” occurs multiple times – please revise this.

Response 3: We have significantly condensed the Introduction to eliminate the "mini-review" style.

  • Streamlining: We summarized the extensive listing of bioactivities and historical pharmacopoeia details into concise statements.
  • Focus: The discussion on legal protection now focuses strictly on the threatened status in China and Europe.

·       Narrative: We removed repetitive phrasing regarding genetic variation and refined the flow to prioritize specific knowledge gaps, ensuring the text focuses squarely on the "genetics–geography–chemistry" framework.

Comments 4: Materials and Methods (pages 3-6, lines 107-245):

I suggest that you provide clearer structuring and remove redundant details in this segment of your research article.

The Methods section reads as a long technical protocol, especially SSR and SNP workflows.
Some parts appear copied from standard lab manuals, please revise this part of Materials and methods.

Generally, my suggestions include the following recommendations:

  • Summarize PCR conditions and reference established protocols rather than describing every step in detail.
  • Move lengthy primer design parameters (Primer3 settings, PCR volumes, PAGE descriptions) to Supplementary Material.

Response 4: We agree that the original section was overly detailed. We have extensively revised it to focus on experimental strategy rather than step-by-step instructions.

  • Removal of Details: Routine descriptions (e.g., pipetting volumes, thermal cycling steps) were removed.

·       Supplementary Material: Extensive technical parameters have been moved to the Supplementary Materials, including MISA criteria (Table S2), Primer3 parameters (Table S3), PCR conditions (Table S4), SNP filtering criteria (Table S5), HPLC Gradient (Table S6), and Reference Standards (Table S7).

Comments 5: Materials and Methods (pages 3-6, lines 107-245):

  • Clarify why only 47 of 53 samples had sclerotia and had metabolite profiling, and how this uneven sampling may influence chemotype interpretation.

Response 5: We have revised Section 4.4 to explicitly state that six laboratory-preserved strains were excluded due to the biological absence of sclerotia.

  • No Bias: This exclusion was based solely on tissue availability rather than chemical selection, making the missing data random with respect to chemotype.

·       Representation: The remaining 47 samples cover all six genetic clades (including Groups 1 and 6). Therefore, we are confident this does not introduce systematic bias into the lineage-specific metabolite analysis.

Comments 6: Results (pages 6-24, lines 246-515)

You included excessive figures and some of them need consolidation.

In general, there are too many figures for the same result type (multiple PCA plots, multiple structure plots).

To be revised:

  • PCA figures for SSR and SNP appear redundant.
  • STRUCTURE and ADMIXTURE outputs could be merged.
  • Some dendrograms are difficult to read due to small fonts, especially colored branches.

In general, I want to recommend following:
Combine related figures:

  • Merge PCA results into one panel (SSR + SNP).
  • Merge STRUCTURE and ADMIXTURE outputs for both marker types.
  • Improve font size, resize labels, and simplify color schemes.

 Response 6: We have consolidated the figures to improve logical flow:

  • PCA/PCoA: Results for both marker types have been merged into a single multi-panel figure (Figure 2).
  • Population Structure: STRUCTURE and ADMIXTURE plots have been combined into Figure 3 for direct comparison.

·       Dendrograms: UPGMA dendrograms (Figure 4) have been redrawn with larger fonts and distinct colors for clarity.

Comments 7: Results (pages 6-24, lines 246-515)

Regarding statistical analysis of chemical data requires clarification, because only four compounds were measured; interpretation of “chemotype clusters” may be overstated.

Recommendation:
You could include:

  • Normality tests (Shapiro–Wilk or Kolmogorov–Smirnov)
  • Post hoc tests (Dunn’s or Tukey)
  • Effect sizes (η² or r)
  • A PCA of chemical data could provide stronger chemotype separation.

 Response 7: We sincerely appreciate these suggestions and have rigorously revised the statistical analysis:

            •          Terminology: We replaced broad terms with precise descriptions like "accumulation patterns of target components."

            •          Tests: We performed normality tests (Shapiro-Wilk).

o   For normal data (polysaccharide/ergosterol), we used One-way ANOVA with Tukey HSD (η²).

o   For non-normal data (polyporusterones A/B), we used the Kruskal-Wallis test with Dunn’s post hoc (ϵ²).

            •          PCA Biplot: A PCA biplot has been added to the Results. It visualizes the distinct separation of component profiles driven by genetic lineage, strongly supporting our quantitative findings.

Comments 8: Discussion (pages 24-28, lines 516-711)

This segment is too long and partially repetitive. It restates the results across multiple paragraphs, which affects clarity and readability. I recommend shortening and streamlining it to avoid redundancy. Also, avoid citing too many references when not necessary (e.g., rDNA evolution discussion), and please clearly separate “genotype–chemotype relations” from “geographic structure.”

I suggest to you:

  • Remove sections that repeat the Results almost verbatim.
  • Reduce speculative statements regarding origin, historical dispersal, and domestication unless literature supports these claims.
  • Shorten section 4.1 and 4.2, which re-explain SSR/SNP methods.

Response 8: We have extensively revised the Discussion for conciseness:

  • Streamlining: We shortened Sections 4.1 and 4.2 by removing unnecessary methodological comparisons.
  • Content: We removed verbatim repetition of results and excised speculative statements regarding historical dispersal.

·       Structure: The text is now reorganized to clearly differentiate the discussion of "Genetic and Geographic Structure" from "Genotype–Chemotype Relationships."

Comments 9: Conclusions (pages 28-29, lines 712-728)

The paper ends abruptly without a structured conclusion. Could you please add a concise Conclusions summarizing six clades established, Qinling–Daba as putative diversity center, Clear genotype–chemotype link, and at the end implications for breeding, conservation, and pharmacopoeial standardization

 Response 9: We have rewritten the Conclusions to provide a structured summary covering the four requested pillars:

  • The delineation of six genetic clades.
  • Identification of the Qinling–Daba Mountains as the diversity center.
  • Clarification of the genotype–chemotype link (identifying superior germplasm).

·       Implications for urgent conservation and pharmacopoeial standardization.

Comments 10: Minor comments regarding aubmitted Figures:

  • UPGMA dendrogram: labels too small; colors too similar.
  • STRUCTURE barplot: consider grouping by geographic region.
  • ML tree: colors are good, but bootstrap values should be more visible.

Response 10: We have revised the figures as follows:

·       UPGMA: We increased label font size, applied bold formatting, and adjusted color saturation for visual distinction.

·       ML Tree: Bootstrap values are now larger, bolded, and high-contrast.

·       STRUCTURE Grouping: regarding the suggestion to group by geography: We have maintained grouping by genetic cluster (K=6) for consistency. Our Mantel test ($r=0.38$) indicates only a modest correlation between genetic and geographic distance, likely due to human-mediated transport mixing genotypes within localities. Therefore, grouping by genetic cluster provides a more accurate visualization of the true population structure than geographic grouping.

3. Response to Comments on the Quality of English Language

Response: The manuscript has been professionally edited by a native English-speaking team to ensure the language meets academic standards.

Reviewer 3 Report

This is an interesting and high-impact original manuscript providing the first comprehensive germplasm evaluation of Polyporus umbellatus, employing a novel and robust "genetics–geography–chemistry" multidimensional framework. As its main merit, the work fills a major research gap, establishing a foundational platform for developing chemotype- driven, molecular-oriented breeding strategies and enabling the domestication of elite medicinal lines of this traditional Chinese medicinal germplasm.

The research is well-designed, employs state-of-the-art methodologies, and addresses a critical gap in the sustainable utilization of a valuable edible and medicinal mushrooms. The potential applications are significant for both conservation biology and the medicinal mushroom industry.

Although the manuscript presents a relevant contribution to the field of medicinal mycology and fungal population genetics, there are some issues, primarily concerning data presentation, statistical depth, and clarity of writing, that need to be addressed before it is suitable for publication.

This is an interesting and high-impact original manuscript providing the first comprehensive germplasm evaluation of Polyporus umbellatus, employing a novel and robust "genetics–geography–chemistry" multidimensional framework. As its main merit, the work fills a major research gap, establishing a foundational platform for developing chemotype- driven, molecular-oriented breeding strategies and enabling the domestication of elite medicinal lines of this traditional Chinese medicinal germplasm.

The research is well-designed, employs state-of-the-art methodologies, and addresses a critical gap in the sustainable utilization of a valuable edible and medicinal mushrooms. The potential applications are significant for both conservation biology and the medicinal mushroom industry.

Although the manuscript presents a relevant contribution to the field of medicinal mycology and fungal population genetics, there are some issues, primarily concerning data presentation, statistical depth, and clarity of writing, that need to be addressed before it is suitable for publication.

Considering the balance between strengths and weaknesses, I suggest a minor revision of the manuscript. The main points for revision are:

  • The methodological approach is a key strength of the manuscript. I recommend to revise these two aspects:
  1. Statistical Analysis of Chemical Data: The use of non-parametric tests (Kruskal-Wallis, Mann-Whitney U) is noted, but the description is sometimes unclear. For instance, when stating "Groups 1, 2 and 5 surpassed Group 6 (P < 0.05)", it should be specified if this is based on post-hoc pairwise comparisons following the test. The details of these post-hoc tests should be clearly stated.
  2. Mantel Test Interpretation: The authors correctly report a weak but significant genetic-chemical distance correlation (r=0.09). However, the discussion should more explicitly address what this low R-value means. It suggests that while genetic lineage is a significant predictor, a large portion of the chemical variation is due to other factors (e.g., environment, epigenetics, unmeasured genetic factors). This is a crucial nuance.
  3. In Materials and Methods or in 3.6/4.4 (Results/Discussion), I suggest to comment briefly the reasons for choosing total carbohydrates (comprise several sugars in addition to polysaccharides) measured using the phenol-sulfuric acid method instead of a specific polysaccharide fraction, representative of Polyporus umbellatus. This recommendation should be taken into account for future studies.
  • With respect to Results Presentation:
    1. Referencing Figures/Tables: The manuscript is severely hampered by numerous "Error! Reference source not found." placeholders (e.g., in Sections 3.1, 3.2, 3.3). This must be corrected before review can be finalized.
    2. The results are generally clear, but the text is sometimes repetitive between the SSR, SNP, and phylogenetic sections. Maybe, streamlining could improve readability.
  • The Discussion is comprehensive and effectively contextualizes the findings. While the perspective on conservation and breeding is excellent, a short, dedicated paragraph on the challenges (e.g., difficulty of controlled sexual crosses, long growth cycle of sclerotia, regulatory hurdles for genetically characterized cultivars) would provide a more balanced outlook. Briefly discussing the challenges in implementing the proposed breeding strategies would strengthen the discussion.
  • The conclusions are well-supported by the data presented and they accurately summarize the main findings. The recommendations are logical and directly follow from the results. I recommend to include the implications of the results for developing consistent quality-control markers for food and pharmaceutical industries.
  • The information contained in “Acknowledgement” is more suitable for “Author contributions”

Author Response

1. Summary

Thank you very much for your time and effort in reviewing our manuscript. We truly appreciate your positive assessment of our methodological approach and your insightful suggestions regarding the statistical interpretation, discussion of breeding challenges, and industrial applications.

We have carefully addressed all your comments. Detailed responses are listed below, and all corresponding revisions have been highlighted in the re-submitted files.

2. Point-by-point response to Comments and Suggestions for Authors

Comments 1:

Statistical Analysis of Chemical Data: The use of non-parametric tests (Kruskal-Wallis, Mann-Whitney U) is noted, but the description is sometimes unclear. For instance, when stating "Groups 1, 2 and 5 surpassed Group 6 (P < 0.05)", it should be specified if this is based on post-hoc pairwise comparisons following the test. The details of these post-hoc tests should be clearly stated.

Response 1: Thank you for highlighting this ambiguity. We have significantly clarified the statistical description in Section 3.8:

  • Specific Post-hoc Tests: We now explicitly state that all pairwise comparisons are derived from rigorous post hoc analyses: Tukey HSD for parametric ANOVA and Dunn’s test with Bonferroni correction for non-parametric Kruskal-Wallis tests.

·       Clear Reporting: The revised text includes precise test statistics, effect sizes, and adjusted P-values Padj for all specific group comparisons to ensure transparency and statistical rigor.

Comments 2: The authors correctly report a weak but significant genetic-chemical distance correlation (r=0.09). However, the discussion should more explicitly address what this low R-value means. It suggests that while genetic lineage is a significant predictor, a large portion of the chemical variation is due to other factors (e.g., environment, epigenetics, unmeasured genetic factors). This is a crucial nuance.

Response 2: We appreciate this important insight. We have revised Discussion 4.4 to explicitly interpret the low r-value. We now state that the metabolic phenotype is highly multifactorial, likely heavily influenced by environmental conditions, epigenetic modifications, or unmeasured genetic factors. We contrast this global plasticity with the lineage-specific accumulation patterns observed in the PCA to provide a more nuanced understanding of genotype-chemotype coupling.

Comments 3: In Materials and Methods or in 3.6/4.4 (Results/Discussion), I suggest to comment briefly the reasons for choosing total carbohydrates (comprise several sugars in addition to polysaccharides) measured using the phenol-sulfuric acid method instead of a specific polysaccharide fraction, representative of Polyporus umbellatus. This recommendation should be taken into account for future studies.

Response 3: We fully accept this suggestion. In the revised Discussion 4.5, we have explicitly acknowledged that while the phenol-sulfuric acid method serves as an effective high-throughput screen for total abundance, it fails to capture structural complexity. We have added a perspective emphasizing that future research must prioritize structural characterization (e.g., monosaccharide composition, backbone architecture) to distinguish bioactive differences between morphotypes effectively.

Comments 4: With respect to Results Presentation:

Referencing Figures/Tables: The manuscript is severely hampered by numerous "Error! Reference source not found." placeholders (e.g., in Sections 3.1, 3.2, 3.3). This must be corrected before review can be finalized.

Response 4: We sincerely apologize for these formatting errors, which resulted from broken cross-reference links during the compilation of the manuscript. We have manually updated all cross-references in the revised version to ensure that every figure and table citation is displayed correctly.

Comments 5: The results are generally clear, but the text is sometimes repetitive between the SSR, SNP, and phylogenetic sections. Maybe, streamlining could improve readability.

Response 5: To address the concern regarding redundancy, we have integrated the SSR and SNP population structure results into a unified section (Section 3.3). Instead of describing the grouping patterns twice, the revised narrative now focuses on the high concordance between the two marker systems, highlighting where they align and where SNP data provides superior resolution regarding gene flow.

Comments 6: The Discussion is comprehensive and effectively contextualizes the findings. While the perspective on conservation and breeding is excellent, a short, dedicated paragraph on the challenges (e.g., difficulty of controlled sexual crosses, long growth cycle of sclerotia, regulatory hurdles for genetically characterized cultivars) would provide a more balanced outlook. Briefly discussing the challenges in implementing the proposed breeding strategies would strengthen the discussion.

Response 6: We sincerely appreciate this suggestion to add a balanced perspective. We have added a dedicated paragraph in Section 4.5 to address practical hurdles.

  • Regarding Sexual Crosses: We would like to note that the technical barrier to achieving sexual hybridization has been recently overcome. Our group reported the first successful crossbreeding strategy for P. umbellatus (Li et al., 2024)

·       Revised Focus: Consequently, our revised discussion acknowledges this progress but highlights the subsequent major bottlenecks for implementation: specifically, the prolonged vegetative growth cycle (3–4 years) required to stabilize and screen these newly generated hybrids, and the regulatory challenges for commercializing molecularly distinct cultivars.

Li, S. J., Li, B., Xu, X. L., Liu, Y. Y., Xing, Y. M., & Guo, S. X. (2024). Insight into the nuclear distribution patterns of conidia and the asexual life cycle of Polyporus umbellatus. Fungal Biology, 128(6), 2032–2041. https://doi.org/10.1016/j.funbio.2024.08.001

Comments 7: The conclusions are well-supported by the data presented and they accurately summarize the main findings. The recommendations are logical and directly follow from the results. I recommend to include the implications of the results for developing consistent quality-control markers for food and pharmaceutical industries.

Response 7: We appreciate this specific recommendation. In the revised Conclusions, we have explicitly addressed the implications for the pharmaceutical sector by proposing a stratified quality-control strategy:

  • Validating ergosterol as a consistent baseline marker for biomass stability.

·       Advocating for molecular-assisted selection and polysaccharide structural characterization to replace inconsistent morphological grading.
This approach aims to ensure precise and consistent quality standards for industrial applications.

Comments 8: The information contained in “Acknowledgement” is more suitable for “Author contributions”

Response 8: We agree. We have moved the statement regarding authorship responsibility from the Acknowledgments section to the Author Contributions section.

4. Response to Comments on the Quality of English Language

Response: The manuscript has been professionally edited by a native English-speaking team to ensure the language meets academic standards.

Reviewer 4 Report

The manuscript presents a comprehensive study on Polyporus umbellatus genomic diversity and the interplay between the genotypes, chemotypes, morphotypes and geographic ogirin of the strains. For the first time the six genetic groups of P. umbellatus were identified and it was shown that almost all groups have a distinct chemotype and morphotype. Surprisingly, there was virtually no correlation between morphotypes and chemotypes. The significance of the study is beyond any doubt as it lays the foundation for the selection of P. umbellatus strains for medicinal use as well as for the development of chemotype-driven, molecular-oriented breeding strategies aimed at creating the strains with high adaptability as well as high capacity for the medicinal compounds production. 

What material was used to quantify the medicinal conpounds? Were these sclerotia obtaind from natural sources or were they fungi cultivated in the laboratory? This needs to be clarified. 

How many strains from each group were used for chemotyping?

L369 and below: Do you mean Principal Component Analysis or Principal Coordinate analysis? The generally accepted abbreviation for principal component analysis is PCA, and for principal coordinate analysis is PCoA.

Please, check the figures across the Manuscript. hey appear to be multiple copies of the same figures presented in the text. The references to the figures and tables should also be checked. 

The text "Error! Reference source not found.." appears in several places in the Manuscript. Please add appropriate references if necessary.

The English is OK, but several typos were detected. Please double-check the manuscript.